# An Outbreak of Limping Syndrome Associated with Feline Calicivirus

**DOI:** 10.3390/ani13111778

**Published:** 2023-05-26

**Authors:** Gianvito Lanave, Alessio Buonavoglia, Francesco Pellegrini, Barbara Di Martino, Federica Di Profio, Georgia Diakoudi, Cristiana Catella, Ahmed H. Omar, Violetta I. Vasinioti, Roberta Cardone, Giacinto Santo, Vito Martella, Michele Camero

**Affiliations:** 1Department of Veterinary Medicine, University Aldo Moro of Bari, 70010 Valenzano, Italy; alessio.buonavoglia85@gmail.com (A.B.); francesco.pellegrini@uniba.it (F.P.); georgiadiakoudi@gmail.com (G.D.); cristiana.catella@uniba.it (C.C.); ahmed.omar@uniba.it (A.H.O.); violetta.vasinioti@uniba.it (V.I.V.); roberta.cardone@uniba.it (R.C.); vito.martella@uniba.it (V.M.); michele.camero@uniba.it (M.C.); 2Department of Veterinary Medicine, University of Teramo, 64100 Teramo, Italy; bdimartino@unite.it (B.D.M.); fdiprofio@unite.it (F.D.P.); 3Andriavet Veterinary Clinic, 76123 Andria, Italy; veterinaria.andria@libero.it

**Keywords:** feline calicivirus, cats, limping disease

## Abstract

**Simple Summary:**

Feline calicivirus (FCV) is a highly contagious virus found in cats and a cause of upper respiratory and oral infections. Typical clinical signs of FCV include nasal discharge, gingivitis, and stomatitis. FCV is also able to affect the joints of cats, resulting in lameness. In this study, we monitored a small outbreak of FCV limping disease in two household cats. The transmission between the two animals likely occurred indirectly via virus shed in the environment from the respiratory tract. The findings of this study highlight the need for the adoption of adequate prophylaxis measures to prevent the transmission of highly transmissible infectious diseases.

**Abstract:**

Feline calicivirus (FCV) is a common viral pathogen found in domestic cats. FCV is highly contagious and demonstrates a high genetic variability. Upper respiratory tract disease, oral ulcerations, salivation, and gingivitis–stomatitis have been regarded as typical clinical signs of FCV infection. Ulcerative dermatitis, abortion, severe pneumonia, enteritis, chronic stomatitis, and virulent systemic disease have been reported more sporadically. Limping syndrome has been also described either in naturally or experimentally FCV-infected cats. In this study, we monitored a small outbreak of FCV infection in two household cats, in which limping disease was monitored with a 12-day lag time. The complete genome sequence was determined for the viruses isolated from the oropharyngeal and rectal swabs of the two animals, mapping up to 39 synonymous nucleotide mutations. The four isolates were sensitive to low pH conditions and trypsin treatment, a pattern usually associated with viruses isolated from the upper respiratory tract. Overall, the asynchronous pattern of infections and the results of genome sequencing suggest that a virus of respiratory origin was transmitted between the animals and that the FCV strain was able to retain the limping disease pathotype during the transmission chain, as previously observed in experimental studies with FCV strains associated with lameness.

## 1. Introduction

Feline calicivirus (FCV) belongs to the genus *Vesivirus*, included in the Caliciviridae family [1]. FCV is an icosahedral virus with a positive-sense, single-stranded RNA genome approximately 7.7 kb in length [2]. The FCV genome contains a virus-encoded protein at the 5′ end, a poly-A tail at the 3′ end, and three adjacent open reading frames (ORFs). ORF1 encodes a polyprotein that is proteolytically cleaved into nonstructural proteins, including a viral protease and the RNA-dependent RNA polymerase. ORF2 encodes the capsid precursor protein, which is post-translationally cleaved into the leader capsid protein and the mature capsid protein VP1. ORF3 encodes the putative minor capsid protein VP2 [3,4].

FCV is an important infectious and endemic pathogen and mostly affects cats under one year of age. FCV infection usually causes mild self-limiting clinical manifestations with high morbidity and low mortality [5,6]. Clinical signs can evolve as acute or subacute infection of the upper respiratory tract and oral cavity (stomatitis and oral inflammatory syndrome) [7,8]. FCV infection is rarely associated with ulcerative dermatitis, abortion, severe pneumonia [2], chronic stomatitis, or virulent systemic disease (VSD) [9]. Vaccination is indicated for all cats [5], although the efficacy of FCV vaccines is affected by viral genetic and antigenic variability [10]. Current vaccines are either mono-valent, including only a single FCV strain (F9 or 255), or bi-valent, including two strains (G1 and 431), but they do not seem to cross-protect effectively against all the field strains [11].

Limping syndrome is also a rare outcome of FCV infection. This syndrome has been described either in naturally or experimentally infected kittens [12,13] and older cats [14]. The peculiar clinical signs are stiffness, hyperesthesia, mild joint pain, and muscle soreness. The pathogenesis of this syndrome likely involves immune complexes [15], and FCV has been isolated from affected joints [12]. FCV antigens have been detected in the joints of cats experimentally inoculated with either a field or a vaccine virus [15]. Thickened synovial membranes and the intra-synovial collection of fluid have been observed. Pyrexia is commonly present, and some cats show concurrent signs of respiratory disease with or without oral ulceration [12].

In this paper, we report a small outbreak of limping disease observed in two household cats. The animals underwent asynchronous infection with a 12-day lag time, suggesting sequential infection between the two animals and, therefore, an intrinsic ability of the FCV strain to induce limping.

## 2. Materials and Methods

### 2.1. Clinical Cases and Sample Collection

A 7-month-old female domestic shorthair (DSH) cat #460/20-1 (cat-1) with outdoor access was presented in November 2020 (day 0) to a veterinary clinic in Andria, Apulia region, Italy. Upon clinical examination, the cat showed lameness, with painful joints, fever (41 °C), oral lesions (painful oral ulcers, stomatitis, glossitis), anorexia, depression, and mild respiratory signs. The clinical signs had started 2 days before presentation to the veterinary clinic (days 2 and 1) (Table 1).

On day 9, a cohabiting male DSH cat #460/20-2 (cat-2) of the same litter displayed similar clinical signs. Both cats were unvaccinated. Serum biochemistry for both cats (at day 7 for cat-1 and day 16 for cat-2) revealed remarkable elevations of the serum amyloid-A protein (240.9–257.1 ug/mL; reference interval [RI] 0.1–0.5 ug/mL) and decreased levels of iron in the blood (21–26 ug/dL; RI 55–152 ug/dL). Serum electrophoresis evidenced an increase in alpha1 (2.2–2.6%; RI 0.8–2.0%), alpha2 (24.9–30.0%; RI 8.0–20.0%), and beta globulins (16.7%; RI 7.0–14.0). These data were indicative of acute inflammation.

After treatment with cefovecin (Convenia, Zoetis; 8 mg/kg/day), prednisolone trimethyl acetate (1 mg/kg/day), and fluid therapy, the cats recovered within 15 days.

Oropharyngeal swabs and rectal swabs (OSs and RSs, respectively) were collected on days 0, 7, and 14 from cat-1 and on days 9, 16, and 23 from cat-2. Blood and serum samples were collected during hospitalization on day 7 from cat-1 and day 16 from cat-2 (Table 1).

The collected samples were sent to the infectious diseases Section of the Department of Veterinary Medicine of the University of Bari with a suspected FCV infection based on clinical presentation.

### 2.2. Nucleic Acid Extraction from Samples

The collected swab samples were processed by homogenization at 10% *w*/*v* in Dulbecco’s minimal essential medium (D-MEM). The supernatant was separated by centrifugation at 2500× *g* for 10 min. Nucleic acids were extracted from the supernatant of swab homogenates, sera, and blood samples using the QIAamp^®^ cador^®^ Pathogen Mini Kit (Qiagen S.p.A., Milan, Italy), according to the manufacturer’s instructions.

### 2.3. Molecular Screening for FCV

Reverse transcription (RT) of RNA extracts was carried out using the GeneAmp^®^ RNA PCR kit (Life Technologies Italia Applera Italia, Monza, Italy). FCV was identified using a quantitative RT-PCR (RT-qPCR) assay based on TaqMan technology targeting the ORF1 region of FCV [16]. A total of 10 μL of the cDNA was added to 15 μL of a master reaction mix containing 0.6 μmol/L of each primer (FCV for GTTGGATGAACTACCCGCCAATC; FCV rev: CATATGCGGCTCTGATGGCTTGAAACTG) and 0.1 μmol/L of the probe (FCV- prob: [FAM] -TCGGTGTTTGATTTGGCCTG- [BHQ1]).

To confirm the presence of FCV RNA, the samples were screened by a one-step RT-PCR assay using the SuperScript One-Step RT-PCR kit (Invitrogen, LifeTechnologies, Milan, Italy) and the forward (Cali1 5′-AACCTGCGCTAACGTGCTTA-3′) and reverse (Cali2 5′-CAGTGACAATACACCCAGAAG-3′) primers, which amplified a 926-bp fragment corresponding to the conserved regions A and B of the ORF2 region of FCV [17].

A nested PCR was performed on a 1:10 dilution of the one-step RT-PCR products using the Hot Master Taq DNA Polymerase (Eppendorf) with the forward (Cali3 5′-TGGTGATGATGAATGGGCATC-3′) and reverse (Cali4 5′-ACACCAGAGCCAGAGATAGA-3′) primers, which amplified a 477 bp portion of the ORF2 gene [17].

The PCR amplicons were purified by the Qiaquick PCR Purification Kit (Qiagen GmbH, Hilden, Germany). Samples with sufficient DNA concentrations (>10 ng/μL) were used for Sanger sequencing at Eurofins Genomics (Milano, Italy). Analysis of the sequences was carried out using BLAST (http://www.ncbi.nlm.nih.gov, accessed on 1 April 2023) and FASTA (http://www.ebi.ac.uk/fasta33, accessed on 1 April 2023) web-based tools.

### 2.4. Molecular Screening for Other Pathogens 

The DNA extracts obtained from OS were screened by PCR assay for the feline herpes virus (FeHV) [18]. Screening for the feline leukemia virus (FeLV) and feline immunodeficiency virus (FIV) was carried out with an immunochromatographic assay (Virbac Test Speed DUO FeLV/FIV, Italy) and confirmed using a nested PCR and RT-qPCR, respectively, on blood and serum samples [19,20].

### 2.5. Cells and Virus

Crandell–Rees feline kidney (CRFK) cells were cultured at 37  °C in a 5% CO_2_ atmosphere in D-MEM. The same medium was used for subsequent experiments. The viruses used in this study included FCV strains 460.20-1/ITA/2020 and 460.20-2/ITA/2020.

### 2.6. Virus Isolation

OS and RS samples from cat-1 and cat-2 were immersed in 1.5 mL of D-MEM and centrifuged at 5000× *g* for 5 min. The supernatant was then treated with antibiotics and inoculated onto CRFK cell monolayers.

Viral growth was monitored daily for the onset of the cellular cytopathic effect (cpe) and cell supernatants were tested by RT-qPCR. Samples with evidence of cpe at the first cell passage were subsequently titrated.

### 2.7. Viral Titration 

Ten-fold dilutions (up to 10^−9^) of each supernatant were titrated in quadruplicates in 96-well plates containing CRFK cells. The titer was calculated by the end-point dilution method after 72 h of incubation at 37 °C.

### 2.8. Seroneutralization (SN)

SN assays were performed on the sera of cat-1 and cat-2, collected on days 7 and 16, respectively. Serial 2-fold dilutions of heat-inactivated sera (56 °C for 30 min) were mixed in 96-well microtiter plates with 100 Tissue Culture Infectious Doses (TCID_50_) of homologous strains isolated from cat-1 and cat-2.

After 45 min of contact at 37 °C, 20,000 CRFK cells per well were added. The neutralization titer of the sera was evaluated after 3 days of incubation.

### 2.9. Evaluation of Susceptibility to pH, Trypsin, and Bile Salts

The in vitro susceptibility to low pH, trypsin, and bile salt treatment of FCVs was investigated and compared with control strains, using protocols previously described [21].

### 2.10. Full-Genome Amplification 

The complete viral genomes of the OSs and RSs of strains 460.20-1/ITA/2020 and 460.20-2/ITA/2020 were generated using consensus primers p1277 (GGCCGCCGGGTTATTGTAAAAGAAATTTGAGACAA) and p1278 (CCGAAGTTGGGGGGGTTTTTTTTTTTTTTTTTTTTTTTTTTCCCTGGGGTTAGGCGCA), binding at the terminations of the FCV genome [21]. Briefly, the RNA was reverse transcribed with primer p1278 using the SuperScript III First-Strand cDNA synthesis kit (Invitrogen Ltd., Milan, Italy). PCR was then performed with TaKaRa LA PCR Kit Ver. 2.1 (Takara Bio, Tokyo, Japan) with primers p1277 and p1278. The amplicons were gel-purified by the Qiaquick Gel Extraction Kit (Qiagen GmbH, Hilden, Germany).

### 2.11. Oxford Nanopore Sequencing

The preparation of the DNA library for sequencing was carried out using the Ligation Sequencing Kit (SQK-LSK110), following the manufacturer’s guidelines. DNA was quantified with the Fluorometric Qubit dsDNA High Sensitivity Assay Kit (Thermo Fisher Scientific, Waltham, MA, USA). Quality control analysis was assessed on the DNA libraries with the High Sensitivity DNA kit of the Agilent 2100 Bioanalyzer (Agilent Technologies, Inc., Santa Clara, CA, USA). Adapters were added prior to library loading on a Flow Cell R9.4.1 and sequencing was performed using a MinION Mk1c sequencer (Oxford Nanopore Technologies, ONT, Oxford, UK). All purification steps were carried out using AMPure XP beads (Agencourt, Beckman Coulter, Brea, CA, USA) according to the SQK-LSK110 sequencing protocol. For the sequencing of the libraries, the NC_12hr_sequencing_FLO-R9_SQK-LSK110 program was run and MinKNOW Software v.4.0.1 (ONT, Oxford, UK) was used for the base calling of raw sequence data.

### 2.12. Sequence and Phylogenetic Analysis

The total paired reads obtained were checked for quality, trimmed, and assembled to reference FCV strains using the Minimap2 plugin implemented in Geneious Prime software version 2022.0.1 (Biomatters Ltd., Auckland, New Zealand).

Full-genome FCV sequences obtained from the NCBI database were aligned using the software Multiple Alignment using the Fast Fourier Transform (MAFFT) plugin implemented in Geneious Prime software version 2022.0.1 (Biomatters Ltd., Auckland, New Zealand). Multiple alignments of the full amino acid (aa) sequence of the capsid and its hypervariable E region were also inspected to identify hallmark mutations based on a review of the literature [22]. The appropriate substitution model settings for the phylogenetic analysis and estimation of selection pressure on coding sequences were derived using “Find the best protein DNA/Protein Models” implemented in the freely available online MEGA X version 10.0.5 software (https://www.megasoftware.net/, accessed on 1 April 2023). The evolutionary history of the nucleotide sequences was inferred by using the maximum likelihood method, six-character states (general time-reversible model), a discrete gamma distribution, and a proportion of invariable sites to model evolutionary rate differences among sites (six categories), supplying statistical support with 1000 replicates.

The evolutionary history of the protein sequences was deduced by implementing a maximum likelihood method and a Jones–Taylor–Thornton (JTT) distance model correct for multiple substitutions based on the model of aa substitution and described as substitution-rate matrices; moreover, a discrete gamma distribution and a proportion of invariable sites were used, providing statistical support with 1000 replicates.

Phylogenetic analyses using other evolutionary models (Bayesian inference, neighbor-joining) were performed to compare the topology of phylogenetic trees. Similar topologies were observed with slight differences in bootstrap values at the nodes of the tree, thus retaining the maximum likelihood tree.

### 2.13. Analysis of the Hypervariable Region E

Seven remarkable residue positions of the hypervariable region E in the capsid region were previously found to be statistically significant for pathotype differentiation [22]. The aa positions were 438, 440, 448, 452, and 455 in the N-HV part of region E, 465 in the central conserved part, and 492 in the C-HV part of region E. Each aa could be described by a set of nine properties, i.e., hydrophobic, positive, negative, polar, charged, small, aromatic, aliphatic, and proline, as previously described [22]. Multiple correspondence analysis (MCA) was performed based on 14 peculiar aa properties of the seven residue positions of the hypervariable region E of strains 460.20-1/ITA/2020 and 460.20-2/ITA/2020 identified in this study. These aa properties were compared with those of a set of 61 FCV strains retrieved from the GenBank database: 4 limping strains, 4 vaccine strains, 36 classical oral respiratory disease (ORD) strains, 6 enteric strains, and 11 VSD strains. All strains were identified or isolated between 1958 and 2019 from several countries worldwide (Table 2).

The MCA was carried out using XLSTAT software (Data Analysis and Statistical Solution for Microsoft Excel, Addinsoft, Paris, France 2017).

### 2.14. GenBank Sequence Submission

The complete genomic sequences of the FCV strains 460.20-1/ITA/2020 and 460.20-2/ITA/2020 identified in the OSs and RSs of cats were deposited in GenBank under the accession numbers OP626899, OP626900, OK428795, and OP626901, respectively.

## 3. Results

### 3.1. Molecular Investigation, Virus Isolation, and Titration

The two cats tested positive with the RT-qPCR assay for FCV. In detail, cat-1 presented an increasing viral load in the OS, ranging from 1.4 *×* 10^3^ RNA copies/mL on day 0 to 6.6 *×* 10^4^ RNA copies/mL on day 14. Viral loads in the RS were 8.1 *×* 10^1^ RNA copies/mL on day 0, 3.0 *×* 10^2^ RNA copies/mL on day 7, and 1.0 *×* 10^0^ RNA copies/mL on day 14.

Cat-2 presented an increasing viral load in the OS ranging from 8.1 *×* 10^1^ RNA copies/mL on day 9 to 1.1 *×* 10^3^ RNA copies/mL on day 23, whilst the RS presented a decreasing viral load ranging from 5.81 *×* 10^1^ RNA copies/mL on day 9 to 1.0 *×* 10^0^ RNA copies/mL on day 23 (Figure 1).

Samples that tested positive for FCV by RT-qPCR were subjected to a nested RT-PCR protocol that amplified a short diagnostic genome fragment (477 nucleotides, nt) of the ORF2 region of FCV. Samples that yielded visible PCR products under gel visualization were subjected to Sanger direct sequencing. The sequences from the OSs and RSs of cat-1 and cat-2 shared 98.7–100.0% nucleotide (nt) identities with each other. By FASTA and BLAST analyses, the sequences displayed the highest nt identities (87.0 to 87.2%) to FCV strain Case 9 (GenBank accession no. KP862871). Furthermore, all the samples analyzed in this study tested negative for FeHV, FeLV, and FIV, ruling out mixed infections.

By visual inspection of the CRFK cell monolayers inoculated with the OSs and RSs collected from cat-1 and cat-2, a cpe referable to calicivirus replication was observed after 24 h. FCV was isolated at the first cell passage from the OSs and RSs collected from both the cats on day 0 and day 9, respectively. The OS and RS of cat-1 presented a titer of 3.25 and 1.75 log_10_ TCID_50_/50 µL, respectively, whilst the OS and RS of cat-2 displayed a titer of 1.50 and 1.75 log_10_ TCID_50_/50 µL, respectively. A second passage on CRFK cells was performed and virus titers increased to 7.0–7.25 log_10_ TCID_50_/50 µL. The viruses were stored at −80 *°C* and then used for the evaluation of phenotypes and for SN.

### 3.2. Sequence Analysis of FCV 

The 7687-nt complete genome sequences of the strains 460.20-1/ITA/2020 and 460.20-2/ITA/2020 were reconstructed from the OSs and RSs of cats. Sequence analyses predicted three ORFs in the genomic sequence of the strains by comparison with other FCVs retrieved from the GenBank database. ORF1 was 5292 nt long (nt 16–5307) and encoded a polyprotein of 1763 aa. ORF2 was 2007 nt in length (nt 5310–7316) and encoded for a capsid protein of 668 aa. ORF3 was 321 nt long (nt 7313–7633) and encoded a protein of 106 aa. Strains 460.20-1/ITA/2020 to 460.20-2/ITA/2020 collected from OSs and RSs shared a 99.5–100% nt identity to each other at the full-genome level, whilst the identity of the deduced aa was 100% for polyproteins and the capsid proteins VP1 and VP2. The strains identified in this study displayed the highest nt identity (equal to 81.3%) to the FCV isolate CH-JL4 (KT206207) in the full-genome; this was followed by the FCV strain GXNN01-19 (MZ712023) in the ORF1 portion (82.6%), the FCV strain 182/2015/ITA (MT008247) in the ORF2 portion (80.6%), and the FCV strain KP331/2020/TH (MZ064640) in the ORF3 portion (90.8%).

Upon phylogenetic analysis based on the full genome sequence, there were several polyphyletic clades, and the strains 460.20-1/ITA/2020 and 460.20-2/ITA/2020 clustered along with FCVs of ORD, VSD, and enteric pathotypes, although not tightly (Appendix A). The overall nt identity of the full genome sequences of strains 460.20-1/ITA/2020 and 460.20-2/ITA/2020 ranged from 76.0% to 81.3% for other FCV strains.

The complete capsid sequences of FCV strains with different pathotypes were retrieved from the databases and used to perform the phylogenetic analysis. In the tree (Figure 2), the strains identified in this study clustered with FCVs displaying ORD pathotypes. A close genetic relatedness to the lameness-associated FCV strain 2280 was also evidenced (Figure 2). Strains 460.20-1/ITA/2020 and 460.20-2/ITA/2020 displayed an overall aa identity of 83.5–92.2% in the capsid region for the other FCVs used for the phylogenetic analysis.

In the phylogenetic tree based on the hypervariable region E, the FCV strains identified in this study formed a clade with ORD and enteric strains; limping-associated FCV strains 1466-1 and 1466-2, identified in cats with polyarthritis in Northern Italy [23]; and the lameness-associated strain 2280, isolated in 1982 in Canada [13] (Figure 3).

### 3.3. MCA 

An initial MCA was carried out on the aa properties of the matched sequences, considering these properties as categorical variables, regardless of the associated pathotype. The expressed percentage of variance was 26.86% for the first axis and 15.34% for the second axis. On the first axis, MCA was able to discriminate between VSD strains and limping, enteric, vaccine, and ORD strains, although the pathotype was not involved in the set of variables (Figure 4). The abscise values were subjected to an ANOVA (analysis of variance) test: the mean value for the abscises of ORD, limping, enteric, and vaccine sequences was significantly different from the mean value for the abscises of VSD sequences (F-test, *p* < 0.0001), substantiating the segregation of ORD, enteric, limping, and vaccine strains from VSD sequences.

Nevertheless, some ORD, enteric, and limping strains remained undifferentiated (i.e., located near the center of the graph) or closer to VSD strains; a unique VSD strain (strain Georgie) was located along with ORD, limping, enteric, and vaccine strains (Figure 4).

A second MCA was performed by adding the “pathotype” as a further variable to the aa properties. The results confirmed those of the first analysis. The uniformity of both analyses confirmed that the aa properties were relevant factors for discriminating between VSD and non-VSD pathotypes.

On the basis of 14 peculiar aa properties from 7 residues, the limping strain identified in this study displayed 4 properties from 4 residues (438, 448, and 452 in the N-hypervariable part of region E and 465 in the central conserved part) that were significantly associated with the limping pathotype (Table 3).

According to the MCA, the limping FCV strain could display a composition of the following properties: polar aa in position 438, small aa in position 448, small aa in position 452, and non-hydrophobic aa in position 465 (Table 3).

### 3.4. SN

The serum of cat-1 collected on day 7 showed an antibody titer of 1:128; the serum of cat-2 collected on day 16 showed an antibody titer of 1:32.

### 3.5. Evaluation of Susceptibility to pH, Trypsin, and Bile Salts

The results of the in vitro evaluation of the effects of pH, trypsin, and exposure to bile salts are reported in Table 4. In the acid lability test, both strains showed a loss of infectious titer of 4.0 log_10_ TCID_50_. When analyzing susceptibility to trypsin, the two isolates showed a 3.5–4.0 log_10_ TCID_50_ reduction. Exposure to bile salts apparently did not affect virus infectivity, with a reduction ranging from 0 to 0.25 log_10_ TCID_50_ 3.2.

## 4. Discussion

FCV markedly varies genetically, antigenically, and phenotypically as a result of a relentless process of evolution based on the accumulation of punctate mutations, persistent infections, and recombination [24]. The existence/distinction of two distinct FCV disease phenotypes, referred to as classical (or respiratory) and VSD (or hypervirulent) forms or pathotypes has now been largely accepted [25,26]. In addition, FCV may also be associated with other peculiar clinical manifestations, such as enteric disease in cats [21,27,28] and limping syndrome [12,14]. Thus far, it has been difficult to decipher the mechanisms of phenotype variation and find genetic hallmarks of this epiphenomenon [22,29]. However, experimental infections have been able to reproduce the VSD form [30], classical upper tract respiratory disease [13,31], enteritis [27], and limping disease [13], suggesting that different FCV strains have different propensities to cause specific disease syndromes [32].

Although lameness was described with FCV infection as early as 1960, it was largely ignored in subsequent studies until the early 1980s [14]. Lameness has since been reported with a number of FCV strains, including modified live vaccine strains [12,15,33,34,35]. In this study, we monitored a small outbreak of limping disease in two unvaccinated 7-month-old household cats. The animals underwent asynchronous infection with a delay of 12 days. Clinical signs included fever, limping, and oral ulcers, whilst the respiratory signs were mild. Both the animals exhibited antibodies 8–10 days after the onset of the disease. Upon genome analysis of the isolates obtained from the OSs and RSs, the viruses detected in cat-1 and cat-2 were highly similar to each other (≥99.5% nt identity) and all 39 nucleotide mutations were found to be non-effective. Virus 460.20-1 RS/ITA/2020 exhibited the highest number of nt mutations (*n* = 34 for virus 460.20-1 OS/ITA/2020, *n* = 39 for virus 460.20-2 RS/ITA/2020, and *n* = 34 for virus 460.20-2 OS/ITA/2020), whilst the other three isolates differed only by 0–5 nt mutations in the consensus genome sequences. Since there is evidence of some phenotype differences between respiratory and enteric FCV isolates [21], the four isolates obtained from cat-1 and cat-2 were tested in terms of their resistance to pH, trypsin, and bile, revealing a pattern of low resistance to pH and trypsin treatment, which was suggestive of a respiratory disease origin. Finally, the neutralizing antibody titer was 3-fold higher in cat-1 than in cat-2; this was likely due to the fact that cat-2 was sampled at an earlier stage of infection (9 days after the onset of clinical signs for cat-1 versus 7 days after the onset of clinical signs for cat-2). Since the cats were not vaccinated, we hypothesized that the antibody response was due to serum conversion, although this was not assessed with serum samples collected at other time points.

Overall, the chronology and sequence data suggest sequential infection of the two animals, with cat-2 being infected by cat-1, likely with a virus derived from the respiratory tract. Curiously, in both animals, the virus from the enteric tract seemed prone to a higher degree of diversification, whilst the isolates made from the oropharyngeal tract of the two animals 9 days apart were virtually identical in the consensus sequence. As cat-1 was hospitalized and, since then, the two cats were kept separated, and considering that the incubation of FCV can range between 2 and 10 days [36], we hypothesize that the infection of cat-2 occurred by indirect contact, i.e., by exposure to the respiratory virus of cat-1 shed in the household, rather than by direct contact with cat-1, despite the owner of the animals being warned and instructed to properly clean and disinfect the household environment. Due to the lack of a viral envelope, FCV is highly resistant to inactivation [37] and FCV infection can occur through contact via fomites [38]. This hypothesis is consistent with the high genetic conservation of the respiratory isolates of cat-1 and cat-2.

A limitation of our study is that we did not carry out experimental infections in cats to reproduce the observed clinical form with the FCV isolates made from the two animals. Several experiments have been carried out to investigate the limping phenotype using different FCV strains and different routes of infection, thus making it difficult to compare the data from the literature. The FCV strain F65, used in several experiments to study FCV-associated lameness, was isolated from an outbreak of lameness and oral disease in a household of unvaccinated cats [35]. Interestingly, the findings observed in the outbreak of febrile lameness described in our study seem to mirror experimental data observed with strain F65 [12]. Twenty-four hours after the intra-articular inoculation of a group of cats with strain F65, the virus was identified in the oropharyngeal cavity of co-housed animals. Additionally, clinical signs of lameness appeared in three out of four contact cats from 5 to 9 days after infection, and lasted for 1 to 5 days [12], suggesting horizontal transmission by a natural route of infection and conservation of the lameness phenotype. In our study, limping appeared in cat-2 with a delay of 12 days and lasted for 11 days, thus suggesting a transmission chain. This would also suggest that the propensity of some FCV strains to cause lameness is maintained over cycles of infections among animals and can be, as in our case, the prominent clinical sign.

Experimental infections with FCV have explored patterns of infection and the onset of the lameness pathotype after oro-nasal exposure [13]. Oro-nasal infection with a pneumotropic FCV strain (FCV 255) and a lameness-associated isolate (FCV 2280) in kittens failed to induce severe upper respiratory signs, but both FCV isolates caused oral ulcers and lameness. However, oral ulcers were more prevalent, and lameness and depression were more pronounced in animals infected with strain 2280. A decrease in lymphocyte counts was also noted only in kittens infected with strain 2280. Kittens with signs of lameness also had increased blood levels of the alpha-1-acid glycoprotein (a₁-AG), which is used to monitor arthritis in humans [39].

In one study, neither intra-nasal infection with the respiratory strain A4 nor subcutaneous inoculation with the vaccine strain F9 was able to induce lameness in unvaccinated cats or in cats vaccinated twice with strain Webster 2113, with the second dose administered 1 month before starting the experiment. Thickened synovial membranes and minimal histological changes were observed only in three cats, whilst the FCV antigen was identified in 14 joints of five cats inoculated with either the attenuated strain F9 or the field strain A4 7 days before euthanasia, but not in animals inoculated 4 days before euthanasia, although the virus could not be isolated from any joints [15]. These findings, whilst confirming the possibility of the localization of FCV in the joints, also demonstrated that the limping pathotype relies on intrinsic characteristics of the FCV strains.

Although the molecular bases of FCV pathotypes have been deciphered, attempts have been made to identify hallmarks able to differentiate VSD from classical respiratory FCVs. A comparison of the capsid region E sequence, which interacts with cell receptors, has identified seven key residues (at positions 438, 440, 448, 453, 455, 465, and 492) that are seemingly significant for pathotype differentiation [22]. When using MCA with Brunet’s criteria, strain 460.20/ITA/2020 showed residue properties typically found in respiratory FCVs (Figure 4). A similar pattern was also observed for other FCV strains associated with limping syndrome, i.e., 1466-1, 2280, F65, and LLK. A limit of this MCA analysis [22] is that it was originally conceived with the specific purpose of differentiating hypervirulent FCVs from classical ORD strains. Therefore, additional residues potentially involved with other pathotypes were not investigated. Accordingly, MCA was tentatively performed, considering strains with non-VSD and non-ORD pathotypes, i.e., enteric, limping, and vaccine strains. In this analysis, 4 aa residues were found to be significantly associated with the limping pathotype. This finding deserves further evaluation using a wider collection of sequences of limping strains.

Additionally, recent studies based on in vitro recombination have revealed that replacing the ORF3 (p30) of strain 2280 with the p30 of strain F9 can affect virulence in vivo. This has been related to the ability of strain 2280 p30 to downregulate the expression of Interferon Alpha and Beta subunit 1 (IFNAR1) and inhibit the phosphorylation of STAT1 and STAT2 in the JAK-STAT signal transduction pathway [40]. Accordingly, genetic tracts outside the ORF2 of FCV could play a role in the pathogenesis of FCV and influence pathotype expression.

Likewise, in the phylogenetic analysis based on the full-genome nt sequence, on the aa sequences of the capsid and the hypervariable region E, various limping-associated FCV strains were clustered in polyphyletic clades. Noteworthily, however, in the capsid-based phylogenies, our limping-associated FCV strains clustered in a group that also included limping-associated FCV strains from Northern Italy isolated in 2017 [23] and the lame-associated strain 2280 isolated in 1982 [13]. This could suggest a common ancestry and retention of the lameness phenotype among these FCV strains, although the massive genetic heterogeneity and lack of precise metadata for some FCV sequences available in the databases evidently challenge this interpretation of the data. For instance, in our case, lameness was the prominent clinical sign, but fever and oral lesions were also present and cat-1 was initially presented to the hospital seeking orthopedical intervention. In other cases, however, lameness could be perceived as a minor clinical sign and the clinical picture could be attributed to respiratory or systemic forms, thus losing relevant metadata.

## 5. Conclusions

In conclusion, we identified a small outbreak of FCV infection with a predominant limping phenotype in unvaccinated cats. In the outbreak, we observed a transmission chain and, based on genome sequencing, the animal-to-animal transmission likely occurred indirectly, with the virus shed by the respiratory tract via contaminated fomites in the household. These findings, whilst reinforcing the notion that the limping syndrome is a reproducible phenotype peculiar to some FCV strains, also stress the need for the adoption of adequate prophylaxis measures to prevent the transmission of highly transmissible infectious diseases. Since vaccines for FCVs are available, the immunization of susceptible cats should always be considered a priority.

## Figures and Tables

**Figure 1 animals-13-01778-f001:**
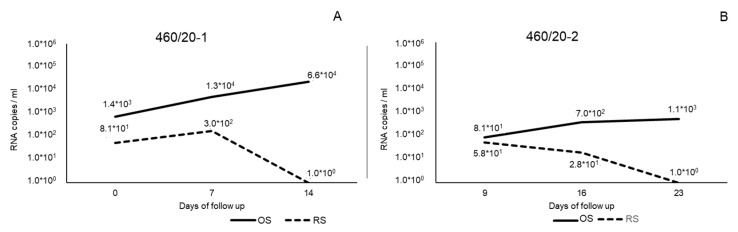
Viral load expressed as RNA copies/mL of FCV strains 460/20-1 (**A**) and 460/20-2 (**B**) identified in this study from oral swabs (OS) and rectal swabs (RS) during days of follow-up.

**Figure 2 animals-13-01778-f002:**
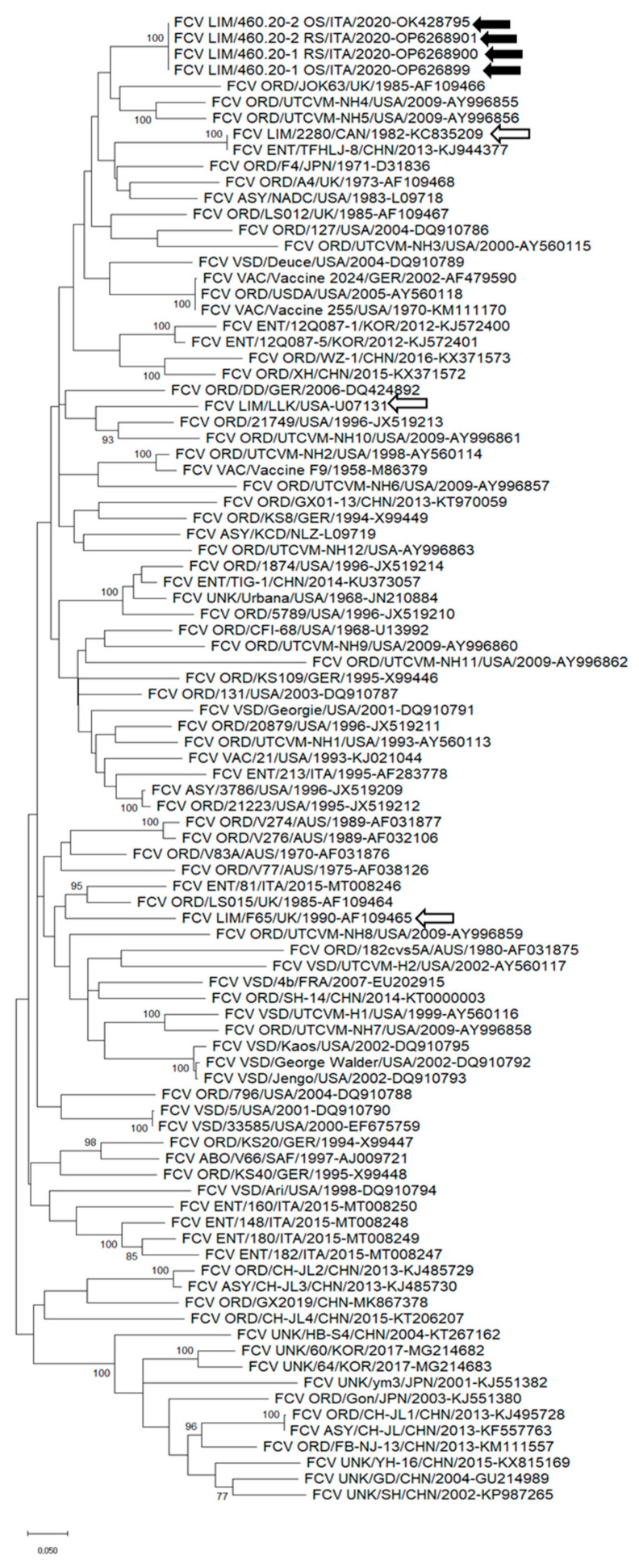
Unrooted phylogenetic tree based on the aa sequence of the full-length capsid proteins of FCV strains detected in this study and cognate strains retrieved from the GenBank database. The tree was generated using maximum likelihood with the Jones–Taylor–Thornton (JTT) distance model, discrete gamma distribution, and a proportion of invariable sites, supplying statistical support with the bootstrapping of 1000 replicates. Black arrows indicate the FCV strains detected in this study. White arrows with black outlines indicate the limping strains retrieved from the GenBank databases. Abbreviations: LIM = limping pathotype; VAC = vaccine; ORD = oral respiratory disease; ENT = enteric pathotype; VSD = virulent systemic disease; CYS = cystitis pathotype; ABO = abortion pathotype; ASY = asymptomatic pathotype; UNK = unknown pathotype.

**Figure 3 animals-13-01778-f003:**
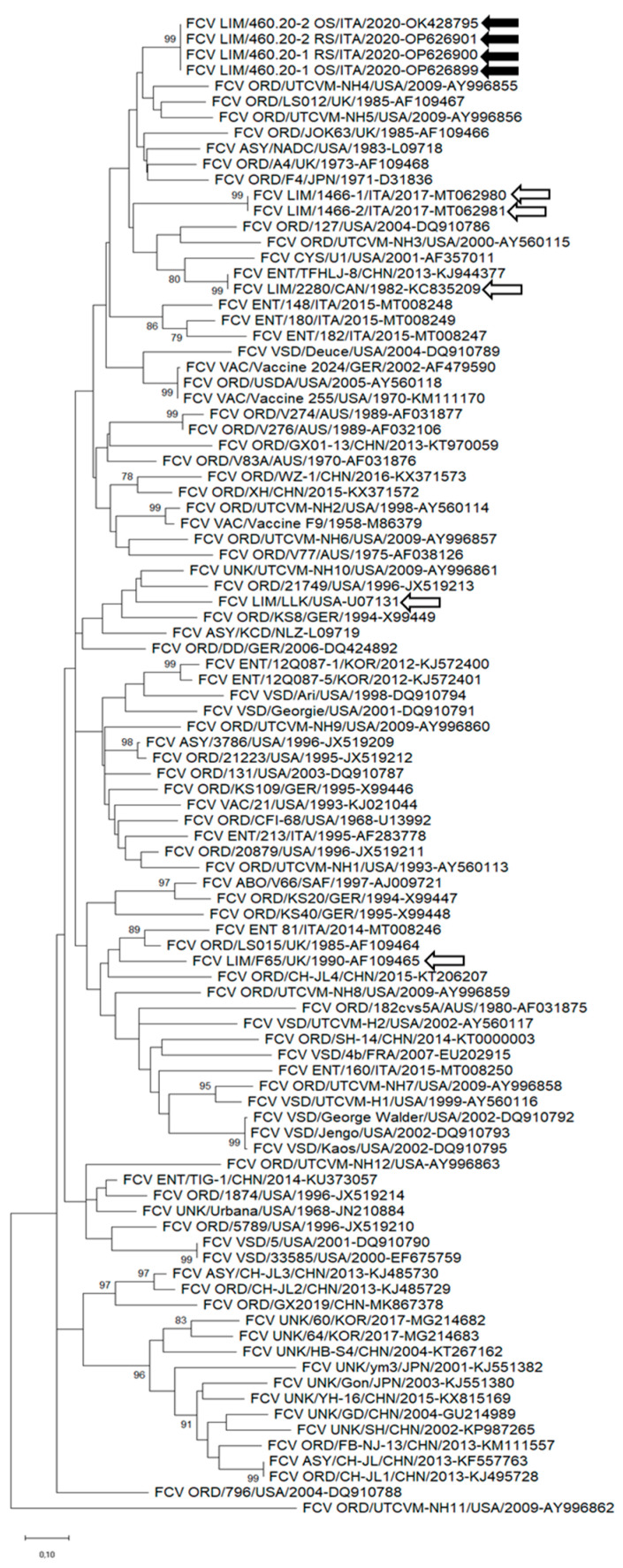
Unrooted phylogenetic tree based on the aa sequence of the hypervariable region E of FCV strains detected in this study and cognate strains retrieved from the GenBank database. The tree was generated using maximum likelihood with the Jones–Taylor–Thornton (JTT) distance model, discrete gamma distribution, and a proportion of invariable sites, supplying statistical support with the bootstrapping of 1000 replicates. Black arrows indicate the FCV strains detected in this study. White arrows with black outlines indicate the limping strains retrieved from the GenBank databases. Abbreviations: LIM = limping pathotype; VAC = vaccine; ORD = oral respiratory disease; ENT = enteric pathotype; VSD = virulent systemic disease; CYS = cystitis pathotype; ABO = abortion pathotype; ASY = asymptomatic pathotype; UNK = unknown pathotype.

**Figure 4 animals-13-01778-f004:**
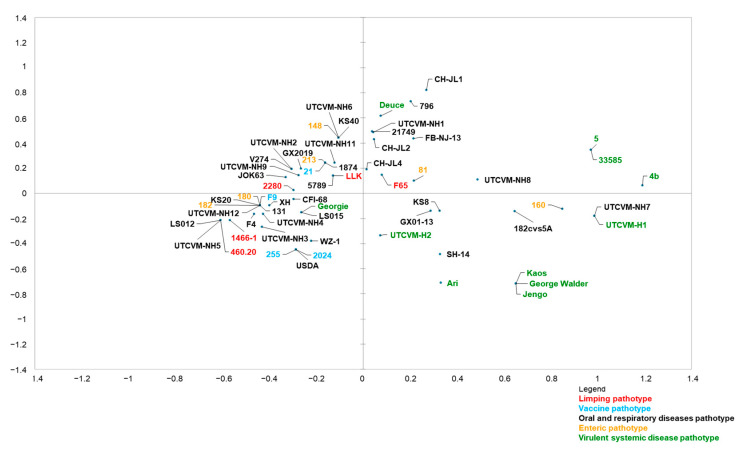
MCA graph. The graph presents the sequences sorted after MCA analysis on the basis of the 14 residue properties of the region E domain of the capsid of strains 460.20/ITA/2020 identified in this study. The horizontal axis differentiates limping (red), oral respiratory disease (black), vaccine (blue), enteric (orange), and virulent systemic disease (green) strains.

**Table 1 animals-13-01778-t001:** Clinical signs and samples collected from the FCV-infected cats. Grey color indicates the viruses that were isolated and sequenced.

Cat	Clinical Signs	Days of Follow Up																										
		−2	−1	0	1	2	3	4	5	6	7	8	9	10	11	12	13	14	15	16	17	18	19	20	21	22	23	24
460/20-1	Lameness	•	•	•	•	•	•	•	•	•	•	•	•	•	•	•	•	•	-	-	-	-	-	-	-	-	-	-
	Fever	-	-	•	•	•	-	-	-	-	-	-	-	-	-	-	-	-	-	-	-	-	-	-	-	-	-	-
	Oral lesions	-	-	•	•	•	•	•	•	•	•	•	•	•	•	•	•	•	-	-	-	-	-	-	-	-	-	-
	Respiratory signs	-	•	•	•	•	-	-	-	-	-	-	-	-	-	-	-	-	-	-	-	-	-	-	-	-	-	-
	Oral swab	-	-	T	-	-	-	-	-	-	T	-	-	-	-	-	-	T	-	-	-	-	-	-	-	-	-	-
	Rectal swab	-	-	T	-	-	-	-	-	-	T	-	-	-	-	-	-	T	-	-	-	-	-	-	-	-	-	-
	Blood and serum	-	-	-	-	-	-	-	-	-	T	-	-	-	-	-	-	-	-	-	-	-	-	-	-	-	-	-
	Hospitalization	-	-	▲	▲	▲	▲	▲	▲	▲	▲	▲	▲	▲	▲	▲	▲	▲	-	-	-	-	-	-	-	-	-	-
460/20-2	Lameness	-	-	-	-	-	-	-	-	-	-	-	•	•	•	•	•	•	•	•	•	•	•	-	-	-	-	-
	Fever	-	-	-	-	-	-	-	-	-	-	-	•	•	-	-	-	-	-	-	-	-	-	-	-	-	-	-
	Oral lesions	-	-	-	-	-	-	-	-	-	-	-	•	•	•	•	•	•	•	•	•	•	•	•	•	•	•	•
	Respiratory signs	-	-	-	-	-	-	-	-	-	-	-	•	-	-	-	-	-	-	-	-	-	-	-	-	-	-	-
	Oral swab	-	-	-	-	-	-	-	-	-	-	-	T	-	-	-	-	-	-	T	-	-	-	-	-	-	T	-
	Rectal swab	-	-	-	-	-	-	-	-	-	-	-	T	-	-	-	-	-	-	T	-	-	-	-	-	-	T	-
	Blood and serum	-	-	-	-	-	-	-	-	-	-	-	-	-	-	-	-	-	-	T	-	-	-	-	-	-	-	-
	Hospitalization	-	-	-	-	-	-	-	-	-	-	-	▲	▲	▲	▲	▲	▲	▲	▲	▲	▲	▲	▲	▲	▲	▲	▲

• presence of symptoms, -: absence of symptoms; ▲: day of hospitalization, T sample collection.

**Table 2 animals-13-01778-t002:** Characteristics of the sequences used for the alignment of the hypervariable region E of the capsid of FCV.

Strain	Pathotype	Country	Year	Access Number
1466-1	LIMP	Italy	2017	MT062980
2280	LIMP	Canada	1982	KC835209
F65	LIMP	United Kingdom	1990	AF109465
LLK	LIMP	Canada	1982	U07131
21	VAC	USA	1993	KJ021044
255	VAC	USA	1970	KM111170
2024	VAC	Germany	2002	AF479590
F9	VAC	USA	1958	M86379
131	ORD	USA	2003	DQ910787
1874	ORD	USA	1996	JX519214
CFI-68	ORD	USA	1968	U13992
CH-JL1	ORD	China	2013	KJ495728
CH-JL2	ORD	China	2013	KJ495729
CH-JL4	ORD	China	2015	KT206207
F4	ORD	Japan	1971	D31836
FB-NJ-13	ORD	China	2013	KM111557
GX01-13	ORD	China	2013	KT970059
GX2019	ORD	China	2019	MK867378
JOK63	ORD	United Kingdom	1985	AF109466
KS40	ORD	Germany	1995	X99448
SH-14	ORD	China	2014	KT000003
USDA	ORD	USA	2005	AY560118
UTCVM-NH2	ORD	USA	1998	AY560114
UTCVM-NH3	ORD	USA	2000	AY560115
UTCVM-NH4	ORD	USA	2009	AY996855
UTCVM-NH5	ORD	USA	2009	AY996856
UTCVM-NH6	ORD	USA	2009	AY996857
UTCVM-NH7	ORD	USA	2009	AY996858
UTCVM-NH8	ORD	USA	2009	AY996859
UTCVM-NH9	ORD	USA	2009	AY996860
UTCVM-NH11	ORD	USA	2009	AY996862
UTCVM-NH12	ORD	USA	2009	AY996863
V274	ORD	Australia	1989	AF031877
WZ-1	ORD	China	2016	KX371573
XH	ORD	China	2015	KX371572
182cvs5A	ORD	Australia	1980	AF031875
796	ORD	USA	2004	DQ910788
5789	ORD	USA	1996	JX519210
21749	ORD	USA	1996	JX519213
KS8	ORD	Germany	1994	X99449
KS20	ORD	Germany	1994	X99447
LS012	ORD	United Kingdom	1985	AF109467
LS015	ORD	United Kingdom	1985	AF109464
UTCVM-NH1	ORD	USA	1993	AY560113
81	ENT	Italy	2015	MT008246
148	ENT	Italy	2015	MT008248
160	ENT	Italy	2015	MT008250
180	ENT	Italy	2015	MT008249
182	ENT	Italy	2015	MT008247
213	ENT	Italy	1995	AF283778
4b	VSD	France	2007	EU202915
5	VSD	USA	2001	DQ910790
33585	VSD	USA	2000	EF675759
Ari	VSD	USA	1998	DQ910794
Deuce	VSD	USA	2004	DQ910789
George Walder	VSD	USA	2002	DQ910792
Georgie	VSD	USA	2001	DQ910791
Jengo	VSD	USA	2002	DQ910793
Kaos	VSD	USA	2002	DQ910795
UTCVM-H1	VSD	USA	1999	AY560116
UTCVM-H2	VSD	USA	2002	AY560117

**Table 3 animals-13-01778-t003:** Correlation table. The variables are recorded as aa position on the capsid protein and the associated property. For each of the properties, the preferred profile of the limping (LIM) strain is indicated (Y = yes; N = no).

Variable	LIM Strain	*p*-Value
438_Hydrophobic	Y	>0.05
438_Polar	Y	0.00325
438_Small	Y	>0.05
440_Hydrophobic	N	>0.05
440_Small	Y	>0.05
448_Hydrophobic	Y	>0.05
448_Small	Y	0.004
452_Negative	Y	>0.05
452_Small	Y	0.0002
455_Negative	Y	>0.05
455_Small	Y	>0.05
465_Hydrophobic	N	0.0015
465_Small	Y	>0.05
492_Positive	N	>0.05

**Table 4 animals-13-01778-t004:** Effects of low pH, trypsin, and bile salts on infectivity of FCVs identified in this study.

FCV Strain	Log_10_ Reduction of FCV Titer (log_10_ TCID_50_) Resulting from Control Virus and Treatment Indicated		
	HCL (pH 3.0)	Trypsin (0.5%)	Bile (0.5%)
	Δ	Δ	Δ
460.20-1 OS/ITA/2020	4.0	4.5	0.0
460.20-1-RS/ITA/2020	4.0	4.0	0.0
460.20-2 OS/ITA/2020	4.0	3.5	0.25
460.20-2 RS/ITA/2020	4.0	4.0	0.0

Note: Resistant phenotypes (null or modest decrease in the infectious titer) are indicated by grey shading. Abbreviations: OS, oropharyngeal swab; RS, rectal swab; Δ = log_10_ reduction.

## Data Availability

The data that support the findings of this study are available from the corresponding author upon reasonable request.

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
