# Peer review of "An Outbreak of Limping Syndrome Associated with Feline Calicivirus"

_animals, 2023, doi:10.3390/ani13111778_

Round 1

Reviewer 1 Report

Dear authors, please consider the following suggestions. 

Line 22 and 533, the word "diffusion" in biological term is defined as the net movement of molecules from higher concentration to lower concentration. Therefore, using this word here is inappropriate. Please consider changing this line to "to prevent transmission of a highly infectious disease".

Table 1: It is confusing for readers to put "H" as day of hospitalisation and "E" for end of hospitalisation on the row for "Lameness". I would suggest to add another row and highlight the period/ duration of hospitalisation for both cats.

Line 100: Can authors please discuss the justification for using cefovecin and prednisolone in these cases? 

Line 453- 454: Please avoid an one sentence paragraph. I would suggest to combine this line with the following paragraph.

Line 481: delete former

Author Response

Dear Referee,

herein you can find a point-by-point response to the comments for the manuscript animals-2384092 entitled “An outbreak of limping syndrome associated with feline calicivirus” submitted to Animals.

Best Regards

R1.1 Line 22 and 533, the word "diffusion" in biological term is defined as the net movement of molecules from higher concentration to lower concentration. Therefore, using this word here is inappropriate. Please consider changing this line to "to prevent transmission of a highly infectious disease".

Reply to R1.1 We thank the referee for the suggestion. We used the suggested terminology “transmission” instead of diffusion in the text.

R1.2 Table 1: It is confusing for readers to put "H" as day of hospitalisation and "E" for end of hospitalisation on the row for "Lameness". I would suggest to add another row and highlight the period/ duration of hospitalisation for both cats.

Reply to R1.2 We thank the referee for the advice. We removed the letter “H” and "E" on the row for "Lameness" and we added a new row “Hospitalization” for both animals highlighting the days of hospitalization.

R1.3 Line 100: Can authors please discuss the justification for using cefovecin and prednisolone in these cases? 

Reply to R1.3 The therapy was prescribed by the clinician and we did not interfere with the indications provided to the owner of the animals. The rationale for the use of prednisolone and antibiotics (cefovecin) was to decrease the pain and inflammation of the joints and to minimize respiratory signs. We agree that the therapy could be modified or planned differently.

R1.4 Line 453- 454: Please avoid a one sentence paragraph. I would suggest to combine this line with the following paragraph.

Reply to R1.4 We joined the two paragraphs, as suggested.

R1.5 Line 481: delete former

Reply to R1.5 We deleted it, as requested.

Reviewer 2 Report

First of all, I appreciate the opportunity to review this manuscript and the thoughtful consideration of this interesting research. These data will be for sure an useful contribution to the scientific literature in both virology and infectious diseases in cats. Please consider the following points to strengthen the manuscript.

1) The two cats included in the study were in contact with other cats (mother, others from the same litter), and if so, do we have information from those animals (were they vaccinated, were they also infected...)

2) Line 131 - You talk about  926bp fragment as a result of a RT-PCR, I personally think that is a really big fragment to be obtained from this techniche. Did you tested or have bibliography that confirms the efficacy of those primers and the correct production of the amplicon?

3) In the lines 195-196 you present a primer that is too long and that also have a lot of bases' repetitions, did you tested and confirmed its efficacy?

4) Lines 405-416 - I personally think that this part of the discussion should be moved to the introduction section

5) It is important to discuss the fact that FCV vaccines are considered internationally as core vaccines, and try to understand why these cats were still not vaccinated, and again to know if there were other cats around and if those were vaccinated or not

Minor corrections:

- Line 41 - Please use italic for the virus genus

- Line 163 - Please use subscript for the 2 in CO2

- Line 178 - Please use subscript for the 2 in CO2

- Line 181 - Replace "cats-1" with "cat-1"

- Line 190 - Please use italic for "in vitro"

- Line 201 - Delete the word "file"

- Line 293 - Replace "cats-1" with "cat-1"

- Line 393 - Please use italic for "in vitro"

- Line 448 - replace "had been" with "being"

- Line 482 - Please replace "was" with "were"

- Line 506 - Please use italic for "in vitro"

- Line 507 - Please use italic for "in vivo"

I'm not an english native but the manuscript seems to be written really well and is clear and easy to read

Author Response

Dear Referee,

herein you can find a point-by-point response to the comments for the manuscript animals-2384092 entitled “An outbreak of limping syndrome associated with feline calicivirus” submitted to Animals.

Thanks in advance

Best regards

Reviewer #2

First of all, I appreciate the opportunity to review this manuscript and the thoughtful consideration of this interesting research. These data will be for sure a useful contribution to the scientific literature in both virology and infectious diseases in cats. Please consider the following points to strengthen the manuscript.

General reply: We thank the referee for his/her appreciation for the manuscript. The efforts of the co-authors are surely paid back by these considerations.

R2.1 The two cats included in the study were in contact with other cats (mother, others from the same litter), and if so, do we have information from those animals (were they vaccinated, were they also infected?)

Reply to R2.1 The two cats described in this study were living alone and they have outdoor access. As stated in the manuscript, both cats were unvaccinated. We cannot rule out that the two cats could have been in contact with other FCV-infected stray cats.

R2.2 Line 131 - You talk about 926bp fragment as a result of a RT-PCR, I personally think that is a really big fragment to be obtained from this technique. Did you tested or have bibliography that confirms the efficacy of those primers and the correct production of the amplicon?

Reply to R2.2 We agree with the referee that a 926bp fragment produced by RT-PCR assay could be larger than routinary molecular approach. However, the RT-PCR assay performed in this study refers to a previous study cited in the text (Marsilio et al., 2005, reference #17). This molecular approach was largely used in previous studies (56 citations in Scopus).

R2.3 In the lines 195-196 you present a primer that is too long and that also have a lot of bases' repetitions, did you test and confirm its efficacy?

Reply to R2.3 Primers p1277 and p1278 used in this study have been designed and employed in a previous report (Di Martino et al., 2020, reference #21). These primers have been designed to amplify the full genome sequence (about 8 kb) and are stabilized with a tail with restriction sites. The primers have been used in several other studies (14 citations in Scopus).

R2.4 Lines 405-416 - I personally think that this part of the discussion should be moved to the introduction section

Reply to R2.4 As suggested by the referee, this paragraph could be eventually moved from the discussion to the introduction section. However, in our opinion, this would change the flow of the narrative and we should modify substantially the rest of the introduction to adapt the text to the new flow of information. Also, it helps the readers, in the discussion, to have a context of the lameness phenotype described in the following paragraph. Accordingly, we decided to retain the discussion as conceived in the original version, provided that the Editor agrees with our decision. 

R2.5 It is important to discuss the fact that FCV vaccines are considered internationally as core vaccines, and try to understand why these cats were still not vaccinated, and again to know if there were other cats around and if those were vaccinated or not

Reply to R2.5 We agree with the referee that it is useful to understand vaccine coverage and usage. The two cats described in this study were unvaccinated as they had been recently adopted by the owner. Although FCV vaccine is considered as a core vaccine, several animals are not vaccinated, i.e. the vast majority of stray cats.

FCV transmission in the population is maintained for different reasons, including low or null vaccine coverage in stray cats, persistent infections, virus stability in the environment, the possibility of contact between FCV-infected stray cats can and household cats with outdoor life style. Also, the polyvalent vaccines administered to cats are designed in order to cross-protect animals from different field FCV strains. However, the emergence of novel FCV strains non-recognizable by vaccine-elicited immunity could decrease vaccine efficacy and pontentially induce vaccine breakthroughs.

These pieces of information are well-known for veterinaries and virologists and, by the way, we could not extend further the discussion. We added a short sentence in the conclusion to reinforce the notion that FCV core vaccines should be used/adopted more extensively as a priority (see pag 15 lines 534-535).

Minor corrections

R2.6 Line 41 - Please use italic for the virus genus

Reply to R2.6 We corrected it.

R2.7 Line 163 - Please use subscript for the 2 in CO2

Reply to R2.7 This was modified.

R2.8 Line 178 - Please use subscript for the 2 in CO2

Reply to R2.8 This was modified.

R2.9 Line 181 - Replace "cats-1" with "cat-1"

Reply to R2.9 We corrected it.

R2.10 Line 190 - Please use italic for "in vitro"

Reply to R2.10 This was modified.

R2.11 Line 201 - Delete the word "file"

Reply to R2.11 The word “file” was deleted, as requested.

R2.12 Line 293 - Replace "cats-1" with "cat-1"

Reply to R2.12 This was corrected.

R2.13 Line 393 - Please use italic for "in vitro"

Reply to R2.13 This was modified.

R2.14 Line 448 - replace "had been" with "being"

Reply to R2.14 This was corrected.

R2.15 Line 482 - Please replace "was" with "were"

Reply to R2.15 This was modified.

R2.16 Line 506 - Please use italic for "in vitro"

Reply to R2.16 This was corrected.

R2.17 Line 507 - Please use italic for "in vivo"

Reply to R2.17 This was corrected.